# The Outcome of BNT162b2, ChAdOx1-Sand mRNA-1273 Vaccines and Two Boosters: A Prospective Longitudinal Real-World Study

**DOI:** 10.3390/v15020326

**Published:** 2023-01-24

**Authors:** Sanaa M. Kamal, Moheyeldeen Mohamed Naghib, Moataz Daadour, Mansour N. Alsuliman, Ziad G. Alanazi, Abdulaziz Abdullah Basalem, Abdulaziz M. Alaskar, Khaled Saed

**Affiliations:** Department of Internal Medicine, Prince Sattam Bin Abdulaziz University College of Medicine, Al-Kharj 16278, Saudi Arabia

**Keywords:** COVID-19, vaccines, booster vaccines doses, effectiveness, COVID-19-related disease, SARS CoV-2 antibody

## Abstract

To date, the effectiveness of COVID-19 vaccines and booster doses has yet to be evaluated in longitudinal head-to-head studies. This single-center longitudinal study assessed the effectiveness of ChAdOx1 nCoV-19, BNT162b2, and mRNA-1273 vaccines and assessed two BNT162b2 boosters in 1550 participants, of whom 26% had comorbidities. In addition, the SARS-CoV-2 antibody dynamics was monitored. A group of 1500 unvaccinated subjects was included as the controls. The study’s endpoint was the development of virologically-proven COVID-19 cases after vaccine completion, while the secondary endpoint was hospitalizations due to severe COVID-19. Overall, 23 (4.6%), 16 (3%), and 18 (3.8%) participants vaccinated with ChAdOx1 nCoV-19, BNT162b2, and mRNA-1273, respectively, developed COVID-19 after vaccine completion, with an effectiveness of 89%, 92%, and 90%. Ten COVID-19 cases were reported in participants with comorbidities, three of whom were hospitalized. No hospitalizations occurred after boosters. SARS-CoV-2 antibody levels peaked 2–4 weeks after the second vaccine dose but declined after a mean of 28.50 ± 3.48 weeks. Booster doses significantly enhanced antibody responses. Antibody titers ≤ 154 U/mL were associated with a higher risk of COVID-19 emergence. Thus, COVID-19 vaccines effectively reduced COVID-19 and prevented severe disease. The vaccine-induced SARS-CoV-2 antibody responses declined after 28–32 weeks. Booster doses induced significant maintained responses. SARS-CoV-2 antibody levels may help determine the timing and need for vaccine booster doses.

## 1. Introduction

COVID-19, the infection caused by SARS-CoV-2, which belongs to the *Coronaviridae* family, has resulted in devastating global public health and economic crises [1]. As of 15 November 2022, over 636 million confirmed cases and over 6.6 million deaths have been reported globally [2]. The high global transmissibility and *pandemicity* of SARS-CoV-2 are multifactorial. The virus transmission occurs along a spectrum that includes droplet transmission, contact with virus-contaminated surfaces, or possible airborne spread [3,4,5]. Besides symptomatic patients, asymptomatic, pauci-symptomatic, or pre-symptomatic people can also spread infection [5,6]. As SARS-CoV-2 evolves, new variants of concern (VOCs) emerge, such as 1.351 (Beta), B.1.617.2 (Delta), and B.1.1.529 (Omicron). There have been concerns that such variants could increase COVID-19 by escaping from immunity generated through previous infection or vaccination or by inducing more severe disease [7,8,9,10].

The genome of SARS-CoV-2 consists of 30,000 nucleotides which encode four structural proteins. The nucleocapsid (N) protein is responsible for packaging the RNA genome, and the membrane (M) protein is accountable for shaping the virions. In addition, the spike (S) and envelope (E) proteins are critical for virions assembly and release, as well as several non-structural proteins (NSP) [11,12,13,14]. SARS-CoV-2 can evade innate recognition, signaling, IFN induction, and IFN-stimulated genes (ISGs) through the viral proteins that block these pathways [15,16]. Infection by SARS-CoV-2 evokes humoral and cellular immune responses. CD4^+^ and CD8^+^ T cells elicited by SARS-CoV-2 infection target various antigens, including structural and non-structural proteins, and are significantly associated with milder disease [17]. Most infected individuals with mild-to-moderate COVID-19 develop neutralizing antibody responses against the viral spike protein, which persist for several months after infection [18,19].

To date, no antivirals directed against SARS-CoV-2 have demonstrated promising efficacy in treating COVID-19. Antiviral therapy started at the early stages of COVID-19 mainly aims to avoid severe infection complications in high-risk patients [20,21]. Thus, mass COVID-19 vaccination campaigns are critical in controlling and slowing the pandemic. At the time of writing the research, more than 100 vaccines have been developed, and 26 vaccines have been evaluated in phase III clinical trials, according to the World Health Organization (WHO) [22]. Within less than 12 months after the beginning of the pandemic, several research teams rose to the challenge and developed vaccines that protect from SARS-CoV-2. The vaccines evoke immune responses, ideally neutralizing antibodies (NAbs) against the SARS-CoV-2 spike protein. Most of the available vaccines are either mRNA vaccines BNT162b2 (Pfizer/BioNTech) and mRNA-1273 (Moderna) that use genetically engineered RNA to generate a protein that safely prompts an immune response or adenoviral-vectored vaccines such as ChAdOx1 nCoV-19 (Oxford/AstraZeneca), Gam-COVID-Vac (Gamaleya), or Ad26.COV2.S (Johnson and Johnson) which are replication-deficient chimpanzee adenovirus (ChAd) vectored vaccines encoding the SARS-CoV-2 spike protein [23]. As of 15 November 2022, approximately 70% of the world’s population has received at least one dose of a COVID-19 vaccine. To date, 12.9 billion doses have been administered globally [2]. However, the current challenge is making these vaccines available to people worldwide and achieving equity in COVID-19 vaccination [24].

The Kingdom of Saudi Arabia (KSA) has reported 825 million COVID-19 cases and 9435 deaths since the pandemic’s start [25]. COVID-19 mass vaccinations in KSA began in mid*-December 2020*. The BNT162b2 mRNA *was the first vaccine approved by* the Saudi Ministry of Health (MOH) and the Saudi Food and Drug Authority (SFDA), followed by ChAdOx1 nCoV-19 and mRNA-1273. To date, 68 million vaccine doses have been administrated, more than 75% of the population is fully vaccinated, and more than 60% have received one or two booster doses [24]. Besides mass vaccination, the Kingdom imposed strict prophylactic measures, including masking and physical distancing.

To date, very few head-to-head studies compared the outcome of different COVID-19 vaccines, their impact on COVID-19 cases, and their progression to a severe disease requiring hospitalization or resulting in deaths. Therefore, the current study was designed to assess the effectiveness of three vaccines (BNT162b2, mRNA-1273, and ChAdOx1- nCoV-19) against concerned outcomes (RT-PCR confirmed COVI-19 cases after vaccination and boosters and COVID-19-related hospitalizations) in vaccinated versus non-vaccinated persons in real-world settings. In addition, this longitudinal study also evaluated the anti-SARS-CoV-2 antibody levels after the different COVID-19 vaccines to investigate if the antibody titers can prioritize persons for vaccine boosters.

## 2. Materials and Methods

### 2.1. Study Design, Study Participants, and Follow-up

This prospective, longitudinal, single-center, real-world study was conducted at Prince Sattam Bin Abdul Aziz University (PSAU) Hospital and vaccination center, Saudi Arabia, to compare the effectiveness of the three vaccines: ChAdOx1-S vaccine and the mRNA vaccines, BNT162b2 and mRNA-1273. Vaccine effectiveness measures the proportionate reduction in RT-PCR-proven COVID-19 among the vaccinated group after complete vaccination (two doses) with or without booster doses. The study also monitored the dynamics of SARS-CoV-2 antibodies after the various vaccines and booster doses. Men and women (18 years and above) eligible for any studied vaccines were enrolled and followed between March 2021 and September 2022.

In the initial phases of this real-life study, not all COVID-19 vaccines were available. Due to the unavailability of an alternative vaccine, the vaccination of individuals with an allergy to a component of the available COVID-19 vaccine or individuals with severe allergic reactions or anaphylaxis after a previous dose of the available COVID-19 vaccine was postponed until other vaccines were available. Patients with medical conditions that warrant delaying the vaccination were temporarily exempted until resolution, invited to join the study, and followed until they could join. If they agreed and provided informed consent, they were enrolled and followed as a control group.

Vaccinated participants and non-vaccinated control subjects were followed at specific time intervals until the end of the study. Participants (vaccinated) and non-vaccinated (controls) were advised to report to the hospital to get tested for COVID-19 using RT-PCR when they developed any symptoms, even mild, or were in contact with COVID-19-positive patients. Participants who tested positive were considered “***cases”***. In addition, COVID-19-related hospitalizations for participants were monitored and reported.

Enrolled subjects were followed at specific time points, namely two weeks after the first dose, then two, four, six, and eight months after the second vaccine dose and first booster dose, respectively. After the second booster, one or two follow-up visits were scheduled. Follow-up visits included physical examination, the COVID-19 antigen test, and SARS-CoV-2 antibody testing.

All vaccinated participants (active groups) and temporarily unvaccinated individuals (control group) provided written informed consent before vaccination, enrollment in the study, and study-related investigations. The study was conducted according to the standards of medical research, including the International Conference on Harmonization (ICH), the Council for International Organizations of Medical Sciences (CIOMS), and the Declaration of Helsinki. Prince Sattam Bin Abdulaziz University Review Board approved and monitored the study (*Approval number: REC-HSD-103-2021*).

### 2.2. Inclusion, Exclusion Criteria, and Control Group

All adults aged 18 years or older were eligible for inclusion if they consented to enrollment in the study and agreed to attend the follow-up visits and perform the necessary laboratory investigations. Individuals with a proven allergy to any of the components of a COVID-19 vaccine were not vaccinated with the vaccine that contains that component.

Individuals with anaphylaxis to a previous dose of one of the COVID-19 vaccines were not provided the second dose of the same vaccine. Persons with a history of severe recurrent thrombosis, thrombosis with thrombocytopenia (TTS), or heparin-induced thrombocytopenia, previous episodes of capillary leak syndrome, cerebral venous sinus thrombosis, antiphospholipid syndrome with thrombosis not enrolled in the ChAdOx1 nCoV-19 vaccine study arm, and were considered for one of the mRNA vaccines.

Vaccination was postponed, and individuals were designated temporary exemption according to the criteria of the Public Health Authorities in the following conditions:oInflammatory cardiac illness within the past three months; myocarditis or pericarditis; acute rheumatic fever or acute rheumatic heart disease (i.e., with active myocardial inflammation); or acute decompensated heart failure.oVaccination was temporarily deferred up to 4 months after a confirmed infection with COVID-19.oUntil recovery from a severe medical condition (such as major surgery or hospital admission for a serious illness).
oUntil recovery from a multisystem inflammatory syndrome in adults.oThe serious adverse event caused by a previous dose of a COVID-19 vaccine, with no acceptable alternative vaccine available.oTreatment with a monoclonal antibody for COVID-19.oInitially, data were limited about the vaccine’s safety in immunocompromised patients (HIV, patients on immunosuppressant drugs, and immunocompromising diseases). The treating physician and the patient discussed the available safety data before administering the vaccine to evaluate risks and benefits.oIn the early phase of the study, data about the vaccine’s safety in pregnant women were limited. Thus, for pregnant women, particularly those with high-risk pregnancies, vaccination was postponed until delivery.

Control group: Subjects with temporary vaccine exemption were invited to join the study as a control group. If they provided informed consent, they were followed until the cause of exemption was resolved and they received the vaccine.

### 2.3. Assessment of Vaccine Effectiveness and Measurement of the Study Outcomes

This study’s primary endpoint was to compare the effectiveness of each vaccine in preventing RT-PCR-confirmed COVID-19 cases and COVID-19-related hospitalizations *versus* those not vaccinated. The secondary endpoint was to compare and monitor the SARS-Cov-2 antibody responses to the different vaccines. During the study, vaccinated or unvaccinated participants developing fever or respiratory symptoms were tested immediately by COVID-19 RT-PCR. Positive cases were followed through home isolation or hospitalization and beyond.

### 2.4. COVID-19 Vaccines Registration, Documentation, and Management

Saudi Arabia has developed a unique and advanced COVID-19 reporting and vaccine registration system through a smartphone or web-based applications: Sehaty, Anat, and Tawakalna (https://ta.sdaia.gov.sa/ accessed on 25 November 2022), respectively. The web-based platforms are considered central repository systems that include COVID-19 PCR results in COVID-19 vaccine data, including the type of delivered vaccine, date, the timing of vaccines, location of vaccination center, and recorded adverse events. In addition, each vaccinated individual was provided with an electronic health passport with a barcode that verifies a person’s immunization status.

### 2.5. Types, Preparation, Storage, and Administration of Vaccines

-***ChAdOx1 nCoV-19* COVID-19***vaccine***:** The multi-dose vial is stored in cold chain conditions of 2 °C to 8 °C for six months. Once opened, the vial can be used after 6 h at room temperature or 48 h in the refrigerator (2 °C to 8 °C). Vial content does not need to be mixed with a diluent.-***BNT162b2 COVID-19****vaccine* (Pfizer-Biontech, Inc., NY, USA) is provided as multi-dose vials. The vaccine is stored frozen and is thawed before dilution. Vials are thawed either in the refrigerator (2 °C and eight °C (35.6 °F and 46 °F)) or at room temperature (up to 25 °C). The BNT162b2 vaccine vial was removed from the refrigerator and kept at room temperature. It should be diluted within 2 h. The BNT162b2 vaccine is administered intramuscularly in the deltoid muscle after cutting the vial. The diluent is 1.8 mL of 0.9% sodium chloride (normal saline, preservative-free). After dilution, the vial contains five 0.3 mL doses of the vaccine that can be used. The interval between the two doses is 21–28 days-*mRNA-1273***COVID-19***vaccine***:** The multi-dose vial is stored frozen and must be thawed before use either in the refrigerator (2 °C and 8 °C) or at room temperature (8 °C and 25 °C). The vial does not need to be mixed with a diluent. The Moderna vaccine is administered intramuscularly in the deltoid muscle. The interval between the two doses is 21–28 days.

### 2.6. Adverse Events (AEs) Recording and Reporting

All adverse events were reported by the study participants and recorded. Participants with AEs were provided with the appropriate management and advice. Severe and unusual adverse events were registered in a special Ministry of Health platform for the recording of AEs.

### 2.7. Measurement of Anti–SARS-CoV-2 Antibody Levels

Serum was collected using standard sampling tubes or tubes containing separating gel. Neutralizing antibody IgG against the SARS-CoV-2 spike protein subunit (S1) was measured using enzyme-linked immunosorbent assay (ELISA) kits according to the manufacturer’s instructions (Elecsys^®^ Anti-SARS-CoV-2 assay, Roche Diagnostics, NJ, USA). The assay uses a recombinant protein representing the nucleocapsid (N) antigen in a double-antigen sandwich assay format, which favors the detection of high-affinity antibodies against SARS-CoV-2. In addition, Elecsys^®^ Anti-SARS-CoV-2 detects antibody titers, which have been shown to correlate positively with neutralizing antibodies in neutralization assays.

### 2.8. Statistical Analysis

We used descriptive statistics to describe the demographic and clinical data of the participants receiving the BNT162b2, mRNA-1273, or ChAdOx1-nCoV-19 vaccines. As appropriate, a comparison of means between vaccinated participants and controls was carried out using the Analysis of Variance (ANOVA) or Kruskal–Wallis test. A Mann–Whitney test was used for pairwise comparisons as a post hoc or follow-up analysis if the test showed differences between the groups. The tested vaccine’s effectiveness (VE) (percentage reduction in COVID-19 incidence in a vaccinated group compared to a non-vaccinated group) was assessed by assessment of parameters. The relative risk (RR) is the risk of the outcome in the treated group (Y) compared to the risk in the control group. The RR, its standard error, and 95% confidence interval are calculated according to Altman, 1991. Absolute risk reduction (ARR) is the difference in risk between the control group (X) and the treatment group (Y). Relative risk reduction (RRR) is the percent reduction in risk in the treated group (Y) compared to the control group (X). The effectiveness = 1-RR × 100%. The number needed to treat (NNT) is the number of patients that need to be treated to prevent one additional bad outcome. The same approach assessed the rate of COVID-19-related hospitalizations [25]. Receiver operating curve (ROC) analysis was used to evaluate the anti-SARS-CoV-2 antibody test diagnostic ability to discriminate COVID-19 positive and negative subjects and the optimal cutoff values. Statistical analysis was performed using SPSS 20 statistical software.( IBM SPSS Statistics for Windows, Version 23.0. Armonk, NY, USA; IBM. Corp)

## 3. Results

### 3.1. Demographics and Characteristics of Enrolled Subjects

This study was conducted at the Prince Sattam Bin Abdulaziz University Hospital and Vaccination Center, Kharj, Kingdom of Saudi Arabia, between March 2021 and September 2022. Initially, 4145 vaccinees were screened for eligibility for the study. Eligible individuals who provided informed consent (n = 1500) were enrolled and prospectively followed until the end of the study (Figure 1). Participants received one of the COVID-19 vaccines: ChAdOx1 nCoV-19 (Cambridge, UK) (Group A; n = 503), BNT162b2 (BioNTech-Pfizer, Mainz, Germany) (Group B; n = 521), or mRNA-1273, (Moderna, Cambridge, MA, USA) (Group C; n = 476). The study cohort completed the two vaccination doses and received the BNT162b2 vaccine’s first and second booster doses. The participants’ follow-ups ranged between 12 to 18 months after vaccination.

The control group included 1500 individuals who were either ineligible due to a documented allergy to components of both vaccines (95; 6.3%) or developed myocarditis or pericarditis after a dose of **mRNA vaccine (9; 0.6%) or** were temporarily exempted from vaccination as a control group. The causes of temporary exemption were previously confirmed COVID-19 (683; 45.53%), severe allergic reactions, or adverse events caused by a previous dose of a COVID-19 vaccine (ChAdOx1- nCoV-19) (187; 12.47%) or contraindication to a vaccine with no available alternative vaccine (174; 11.6%), severe medical conditions or major surgeries (153; 10.2%), high-risk pregnancy (275; 18.33%), and 172 (11.46%) due to other causes such as patients receiving immunotherapy or monoclonal antibody for COVID-19. The control subjects with temporary exemptions were followed until the cause of postponing vaccination was resolved (3–9 months). By the end of the study, that caused 1089 subjects (78%) to be vaccinated.

The active vaccinated cohort included 767 (51.13%) Saudis and 733 (48.87%) non-Saudis (*p* = 0.02) (Table 1). There were no significant differences in participants’ ages or gender in the three vaccinated groups and the control subjects. Among the vaccinated participants, 1416 (94.4%) did not experience previous natural COVID-19 infection and were vaccine naïve. Eighty-four (5.7%) participants reported previous attacks of COVID-19 4–6 months before enrollment in the study. Among the enrolled subjects, 1114 (74.3%) vaccinated participants reported no diseases or health disorders, while 386 (25.7%) participants had one or more comorbidities. The chronic health disorders included diabetes, hypertension, dyslipidemia, cardiac disease, chronic liver disease, chronic renal disease, hemoglobinopathies, neurologic or psychiatric disorders, autoimmune disorders, and malignancies (Table 1). Among the patients with hemoglobinopathies, 88% and 12% had sickle cell anemia and thalassemia, respectively. The autoimmune diseases included systemic lupus erythematosus (SLE), rheumatoid arthritis, psoriasis, and Hashimoto’s thyroiditis. The participants with malignancies included patients with breast cancer, lymphoma, and prostate cancer who completed their therapeutic approaches. The control group had more comorbidities since patients with advanced chronic diseases were initially temporarily exempted from vaccination.

### 3.2. The Outcome of Vaccination in the Three Groups

The study endpoint was the occurrence of new COVID-19 cases after vaccine completion. The study demonstrated that 23, 16, and 18 participants developed COVID-19 RT-PCR-proven COVID-19 after taking the second dose of ChAdOx1 nCoV-19, *BNT162b2,* and *mRNA-1273* vaccines, respectively;(*p* = 0.369), *versus* 554 non-vaccinated subjects (control group) (*p* < 0.0001). The effectiveness of ChAdOx1-nCoV-19, BNT162b2, and mRNA-1273 vaccines was 89%, 92% (95% CI of 95.03% to 98.14%), and 90% (95% CI of 80.8% to 95.5%.), respectively. A significant difference was observed only between BNT162b2 and ChAdOx1 nCoV-19 (*p* = 0.0026) (Table 2). Similarly, a significant difference in absolute risk reduction (ARR) was detected between COVID-19-infected participants within the vaccinated and unvaccinated groups. Only three vaccinated participants, one patient with SLE, one with colorectal cancer, and an obese individual, developed severe COVID-19-related disease and were hospitalized; however, no mortalities were reported in vaccinated participants. In contrast, 103 (18.6%) unvaccinated control subjects were hospitalized with COVID-19-related complications with three mortalities (*p* < 0.0001).

Although the endpoint was the development of COVID-19 cases after vaccine completion, the study also monitored the COVID-19 instances after the first vaccine dose. As shown in Table 1, there was a significant difference in the COVID-19 cases that developed after the first doses of ChAdOx1 nCoV-19, BNT162b2, and mRNA-1273, with the highest issues developing after ChAdOx1 nCoV-19.

Table 3 summarizes the emergence of COVID-19 cases after vaccine completion and booster doses versus unvaccinated control subjects. COVID-19 infections were significantly higher in unvaccinated individuals, with the highest COVID-19 incidence in patients with autoimmune diseases, malignancies, people with diabetes, particularly those with uncontrolled diabetes, and patients with hemoglobinopathies (Table 3). Among the participants who developed COVID-19 after completing the second dose of the vaccines, 73.7% had comorbidities. The reported COVID-19 incidents were significantly lower in vaccinated *versus* unvaccinated participants with comorbidities. The vaccine effectiveness in patients with diabetes mellitus, obesity, autoimmunity, and malignancies ranged between 80% and 86%, which is lower than VE in participants without chronic diseases and those with other comorbidities. None of those who received booster vaccines developed a severe illness that required hospitalization. Figure 2 is a Kaplan–Meier graph that demonstrates the time to event (the occurrence of COVID-19 cases in the three study groups and control subjects).

### 3.3. Vaccine Adverse Events

Several adverse events were reported after receiving the different COVID-19 vaccines; however, most were not serious (Table 4). The most frequent adverse events included pain at the site of injection (89.5%), fatigue (39.6%), headache (25%), low-grade fever (17.8%), and muscle pain (11.3%). Less frequent adverse events included chills (5.8%), abdominal pain (4%), allergic skin reaction (6.2%), joint pains (3.3%), diarrhea, or vomiting (2.1%). In addition, about 2% of those receiving the vaccines reported palpitations (2%), numbness (1.8%), sleep disorder (1.7%), and anxiety/restlessness (0.5%). Fever > 38 extending beyond three days, headache, and abdominal and joint pain were significantly higher among participants receiving ChAdOx1 nCoV-19 (Table 4). Ten participants (0.66%) with comorbidities (diabetes mellitus, hypertension, obesity, sickle cell disease, cancer) developed serious adverse events, including severe allergic reactions/anaphylaxis, Guillain–Barré syndrome, thrombosis without or with thrombocytopenia, seizures, and myocarditis. Two previously controlled sickle cell disease (SCD) participants developed vaso-occlusive crisis (VOC), and a patient with diabetes and valvular heart disease developed deep venous thrombosis shortly after the ChAdOx1 nCov-19 vaccine. One participant vaccinated with the BNT162b2 vaccine developed Guillain–Barré syndrome one day after vaccination. In addition, one participant developed optic neuritis one day after vaccination, and she responded to corticosteroid therapy.

### 3.4. SARS-CoV-2 Antibody Dynamics

The study demonstrated that the SARS-CoV-2 antibodies were slightly higher in participants vaccinated with either BNT162b2 or mRNA-1273 than ChAdOx1 nCoV-19; however, the difference was insignificant (*p* = 0.716) *(data are not shown).* As shown in Figure 2, SARS-CoV-2 antibodies rose gradually after the first vaccine dose and peaked two to four weeks after the second dose. Although the levels of antibodies remained elevated until six months in the three groups, a gradual decline was detected between 24 to 32 weeks (mean 28.50 ± 3.48 weeks) post-vaccination (Figure 3a). The booster doses resulted in the antibody titers with levels that exceeded those after the second vaccine dose. The ROC curve analysis showed that antibody titers ≤ 154 U/mL were associated with an increased risk of acquiring COVID-19 with a sensitivity and specificity of 98% and 99%, respectively (Figure 3b). Patients with comorbidities tended to have slightly lower humoral responses than healthy participants. Furthermore, antibody titers declined earlier (between 16–28 weeks) in vaccinated participants with comorbidities compared to those without.

## 4. Discussion

This prospective, longitudinal, test-negative, real-world study compared the effectiveness, outcomes, and safety of three COVID-19 vaccines: ChAdOx1 nCoV-19, BNT162b2, and mRNA-1273, and evaluated the role of the vaccine boosters in a broad, representative patient population at different ages and ethnicities in diverse clinical settings, where many patients have multiple comorbidities. The study also measured the anti-SARS-CoV-2 antibodies’ levels at other time points after vaccination. Overall, the study provides real-world evidence that the three tested vaccines conferred significant protection against COVID-19 infections and reduced severe outcomes that require hospitalization and COVID-19-related mortalities. The three vaccines were also well tolerated. However, the anti-SARS-CoV-2 antibody titers and, consequently, COVID-19 vaccine effectiveness waned with time but were regained with the booster doses.

The current study demonstrated that the effectiveness of ChAdOx1- nCoV-19, BNT162b2, and mRNA-1273 was 89%, 92%, and 90%, respectively, where the effectiveness of BNT162b2 was significantly higher than ChAdOx1 nCoV-19; however, no significant difference was detected between ChAdOx1 nCoV-19 and mRNA-1273 vaccines. The effectiveness of the vaccines in the current study was comparable with the efficacy or effectiveness rates reported by previous studies [26,27,28,29]. Although the study’s endpoint was the development of COVID-19 cases following the second vaccine dose, we also monitored patients after the first vaccine dose to investigate the magnitude of protection conferred by such a dose. Although the immune response to the vaccines has not been fully developed, our findings showed that vaccinated participants, irrespective of the type, were at lower risk of developing COVID-19 and disease-related hospitalizations compared to unvaccinated individuals in similar prophylactic settings. However, the COVID-positive cases were significantly higher after the first dose of ChAdOx1 nCoV-19 than those detected after BNT162b2 or mRNA-1273 vaccination. This finding may be attributed to the long interval between the first and second ChAdOx1 nCoV-19 and the changes in the adopted prophylactic measures. Given the first dose’s beneficial effects, distributing the vaccines among those eligible as quickly as possible, after providing for the most vulnerable broad mass vaccination campaigns may be beneficial for controlling the viral spread and severe COVID-19-related morbidities and mortalities in resource-limited countries with irregular vaccine supplies [30].

The current study demonstrated a significant reduction in post-vaccination COVID-19 incidents and severe disease after the second dose of the three vaccines. Thus, the present study results lend further credence to the findings of several studies [28,31,32,33] that showed the critical role of vaccine completion in reducing infection and COVID-19-related severe disease. The confirmed COVID-19 infections detected after vaccine completion were 57/1500 (3%), and none of the cases was associated with severe symptoms in this study cohort. Our findings are comparable to those of a study that showed 39 of 1497 (2.8%) fully vaccinated healthcare workers developed post-vaccination SARS-CoV-2 breakthrough incidents [34]. Vaccine breakthrough infections may occur following several vaccines that may result from viral factors, including the emergence of variants, transmissibility rates, immune evasion, or host determinants such as age, comorbidities, and immune status may result in post-vaccination confirmed infections [35]. This study did not assess the prevailing SARS-CoV-2 variants; however, the impact of comorbidities was investigated.

Few countries provided COVID-19 vaccine boosters. In KSA, two booster doses were administered, the first 6–8 months after vaccine completion and the second eight months after the first booster dose. In KSA, initially, priority was given first to elderly persons and high-risk individuals then boosters were offered on a larger scale. After the first vaccine booster, only 10 minimally symptomatic vaccine breakthroughs were detected. No confirmed COVID-19 infections occurred after the second vaccine booster. The current study also demonstrated that the anti-SARS-CoV-2 titers decline 24–28 weeks after vaccine completion per other reports. However, administering the first and second COVID-19 vaccine boosters was associated with a significant increase in anti-SARS-CoV-2 levels and a subsequent sharp reduction in breakthrough infections. Overall, this study showed that booster doses have significantly reduced breakthrough vaccine infections and the risk of hospitalization or death in participants without and with comorbidities.

This study showed that some comorbidities influenced the vaccines’ effectiveness and odds of acquiring COVID-19 after vaccination. Most post-vaccination symptomatic COVID-19 cases and related hospitalizations were observed in patients with comorbidities. Patients with diabetes mellitus, autoimmune diseases, malignancies, and participants with BMI > 40 showed lower VE and were more susceptible to infections and severe COVID-19 than healthy participants. Our findings and others [36,37] demonstrated that unvaccinated obese participants (BMI > 40) showed high susceptibility to acquiring and developing severe COVID-19 and increased risk of hospitalization. Furthermore, obese participants, particularly those with severe obesity (BMI > 40 kg/m^2^), had lower VE than participants with reasonable BMI. Obesity is prevalent in KSA, as a national survey conducted between 1995 and 2000 found that the overall prevalence of obesity among Saudi adults was 36% [38]. Thus, obese patients are considered a high-risk group that must be prioritized for vaccination. In agreement with previous studies, this study showed increased incidence and severity of COVID-19 in patients with diabetes. Diabetes is a public health problem in Saudi Arabia; the World Health Organization (WHO) reported that Saudi Arabia ranks second highest in the Middle East and is seventh in the world for the rate of diabetes [39]. This study showed that COVID-19 infection occurred in 35% of diabetic unvaccinated control diabetic subjects. Some patients, particularly those with uncontrolled diabetes, developed a severe disease that needed hospitalization despite implementing strict prophylactic measures. Other studies reported similar observations [40,41,42,43,44,45]. Diabetes is a chronic disease associated with several metabolic, vascular, and immunologic abnormalities that can affect our response to pathogens. Hyperglycemia and insulin resistance induce oxidative stress and enhance the production of glycosylation end products (AGEs) and proinflammatory cytokines, which mediate tissue inflammation [46]. The inflammatory process may lead to higher infection susceptibility [46]. Unvaccinated cancer patients and those with autoimmune diseases were found in the current study to be highly susceptible to COVID-19 infection with severe symptoms. One study [47] enrolled 23,266 participants with cancer and 1,784,293 without cancer. A total of 10,404 positive COVID-19 cases were reported. Compared with participants without cancer, those living with cancer had a 60% increased risk of acquiring COVID-19. Cancer patients on current chemotherapy or immunotherapy treatment had a 2.2-fold increased risk of a positive test [47]. The present study showed reasonable vaccine response in patients with comorbidities, including diabetes, obesity, cancer, and autoimmune disease. After the second booster vaccine dose, no vaccine breakthrough was reported in this cohort. The findings of this study and another study [48] stress the benefit of booster vaccine doses in the high-risk population.

The COVID-19 vaccines’ effectiveness demonstrated in this study was coupled with a low risk of adverse events. The adverse events of the three types of vaccines assessed in the current study are comparable to those reported in several studies [28,29,30,31,32,33]. Serious vaccine adverse events, severe allergic reaction/anaphylaxis, Guillain–Barré syndrome, thrombosis, thrombocytopenia, seizures, and myocarditis observed in this study were rare as they occurred in 10 (0.66%) participants, all of whom had comorbidities. Thrombosis without or with thrombocytopenia has been reported after COVID-19 vaccines [49,50]. The severe allergic reactions that occurred shortly after vaccination were promptly managed in the hospital. Two pre-vaccine controlled sickle cell disease (SCD) study participants developed vaso-occlusive crisis (VOC) and thrombosis shortly after AstraZeneca, an observation previously reported by some studies [51,52]. SCD patients vaccinated with BNT162b2, or mRNA-1273 did not suffer post-vaccine VOC. A patient with diabetes and valvular heart disease developed deep venous thrombosis following ChAdOx1 nCov-19 vaccine. One patient vaccinated with ChAdOx1 nCov-19 vaccine developed optic neuritis four days after receiving the vaccine and responded to steroid therapy. Recently, multiple reports were published on the development of optic neuropathy following COVID-19 vaccination [53]. Thus, high-risk patients with hypercoagulable states should be thoroughly investigated before vaccination, and careful selection of the suitable COVID-19 vaccine is recommended. Our study demonstrated myocarditis in two patients following BNT162b2 and ChAdOx1 nCov-19 vaccines, a post-vaccination observation reported by several studies [54,55,56]. Post-SARS-CoV-2 vaccination, Guillain–Barre syndrome occurred in the current study in a diabetic participant vaccinated with the BNT162b2r vaccine; however, the patient responded to steroids [57,58,59]. Collectively, our findings and the results of several studies demonstrated the safety of the COVID-19 vaccines and the rare occurrence of serious adverse events. Thus, the benefits of vaccination against SARS-CoV-2 are substantial and outweigh the associated risks.

In agreement with several studies [60,61,62], SARS-CoV-2 antibodies rose gradually after the first vaccine dose. However, significantly high titers were achieved after the second dose of ChAdOx1 nCoV-19, BNT162b2, and mRNA-1273 vaccines. A gradual decline in antibody titers over time was detected in our cohort between 24 and 32 weeks, an observation reported by several groups [60,61,62,63]. The booster doses evoked high SARS-CoV-2 antibody levels that exceeded those after the second vaccine. Although participants with comorbidities had slightly lower humoral immune responses and showed earlier antibody titer decay, they responded well to the booster vaccines as breakthrough infections were detected after the second booster. In this study cohort, participants with antibody titers ≤154 U/mL were at increased risk of acquiring COVID-19. Establishing a cutoff of SARS-CoV-2 that may predict an increased risk of post-vaccination breakthrough infections may help prioritize persons for COVID-19 boosters, individualizing the timing of boosters and conferring better protection, particularly with patients with comorbidities. Further prospective extensive studies are needed to investigate if monitoring the dynamics of SARS-CoV-2 antibodies SARS-CoV-2 antibodies may play a role in prioritizing the use of a limited supply of vaccines.

This study has several limitations. First, the study has not addressed the impact of vaccines on the various SARS-CoV-2 variants that emerged since the introduction of the COVID-19 vaccines. Diagnosis of post-vaccine COVID-19 was based on participants’ self-reporting of symptoms. Thus, there may be a probability of missing asymptomatic cases. Finally, the study has not assessed vaccine efficacy and safety in individuals younger than 18 and children younger than 12.

In conclusion, the three evaluated COVID-19 vaccines were tolerable, safe, and effective in reducing COVID-19 and preventing severe disease. Furthermore, COVID-19 vaccination showed effectiveness and safety in individuals with comorbidities and those without any underlying medical disorders. In addition, the vaccine-induced SARS-CoV-2 antibody responses gradually decline. Booster vaccine doses induced significant maintained responses. SARS-CoV-2 antibody levels may help determine the timing and need for vaccine booster doses, particularly in patients with comorbidities.

## Figures and Tables

**Figure 1 viruses-15-00326-f001:**
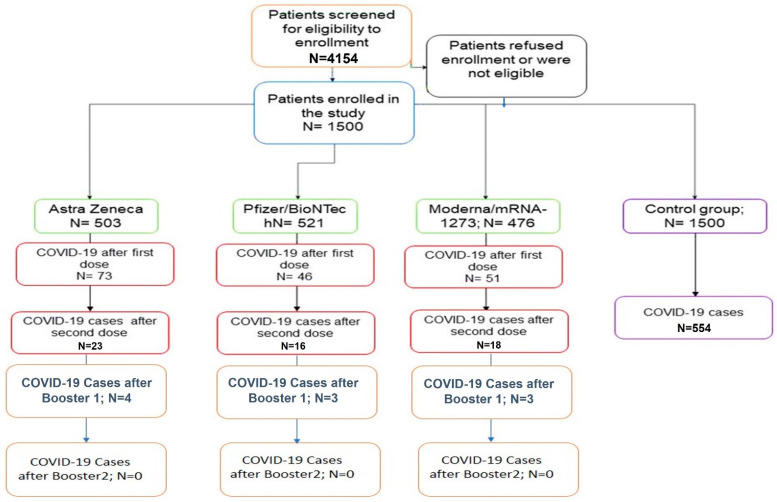
The flow of participants through the study. The figure shows the flow of participants through the study. Initially, 4154 subjects were screened. A total of 3535 were eligible; however, only 1500 subjects agreed to join the study, provided informed consent, and were enrolled. Subjects with temporary vaccine exemption were those recovering from natural COVID-19, those treated with immunosuppressive therapy, and pregnant women.

**Figure 2 viruses-15-00326-f002:**
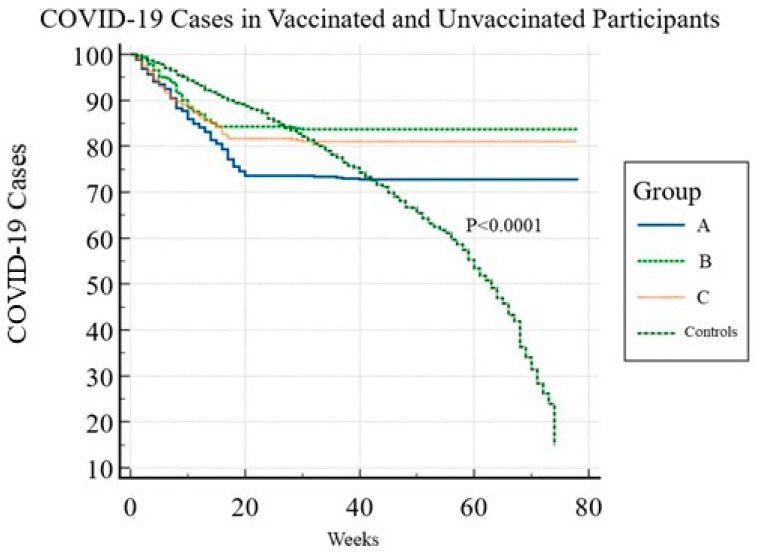
Time to COVID-19 infection in the three study groups and control subjects. Kaplan–Meier survival curves show the time to endpoint (occurrence of COVID-19) in vaccinated and unvaccinated participants throughout the study. In vaccinated participants, more COVID-19 cases occurred after the first dose, given that the post-vaccine immune responses were not wholly developed. After the second vaccine dose, 23 (4.6%), 16 (3.1%), and 18 (3.8%) (*p* = 0.3.69) cases occurred, with 82% of confirmed COVID-19 infections occurring within 5–18 days after the second dose. Seven (0.47%) COVID-19 cases occurred after the first booster dose. No cases were reported after the second booster. A reduction in the rate of confirmed COVID-19 infection was observed over time as compared with the unvaccinated control subjects. The hazard ratios with a 95% confidence interval for the vaccinated groups (Groups A, B, and C) and the unvaccinated controls are shown in the KM hazards table. The hazard ratio measures the magnitude of the difference between the two curves in the Kaplan–Meier plot. In contrast, the *p-value* measures the statistical significance of this difference.

**Figure 3 viruses-15-00326-f003:**
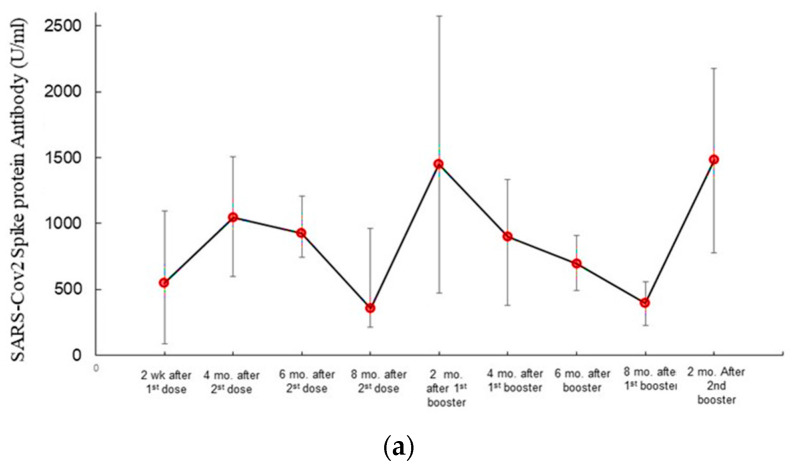
(**a**) SARS-CoV-2 antibody dynamics in vaccinated subjects. (**b**) SARS-CoV-2 antibodies titers in participants without and with post-vaccine COVID-19. (**a**) The figure demonstrates the kinetics of antibody titers against SARS-CoV-2 spike protein during follow-up. A gradual rise in the antibody titers started two weeks after the first vaccine doses. The antibody levels peaked after the second vaccine dose. Antibody levels started to decline 24–32 weeks after the second vaccine SARS-CoV-2 responses. (**b)** The figure shows a significant difference in the SARS-CoV-2 antibody levels in participants who did not develop post-vaccine COVID-19 compared to those who developed COVID-19 after vaccination. A cutoff of 154 U/mL was associated with an increased risk of developing COVID-19.

**Table 1 viruses-15-00326-t001:** Demographics baseline clinical features enrolled subjects according to COVID-19 vaccine type.

Parameters	Group A (n = 503) ChAdOx1 nCoV-19 Vaccine	Group B (n = 521) BNT162b2 Vaccine	Group C (N = 476) mRNA-1273 Vaccine	*p* Value between the Three Vaccinated Groups	Control Group (n = 1500)	*p* Value between Vaccinated and Non-Vaccinated
Age (years) Mean ± SD	46.31 ± 14.99	44.63 ± 15.63	43.79 ± 15.62	0.08	45.09 ± 16.19	0.06
Males; n (%)	295 (58.65)	301 (57.77)	258 (54.2)	0.36	846 (56.4)	0.45
Nationality						
Saudi; n (%)	256 (50.89)	267(51.25)	244 (51.26)	0.89	743 (49.53)	0.38(1.976% to 5.170%)
Non-Saudi; n (%)	247 (49.15)	254 (48.75)	232 (48.75)		757 (50.47)	
Patients with Comorbidities: Total 386 (25.73%)				*p* value between the three vaccinated groups; *p* (95% CI)		*p* value between vaccinated and non-vaccinated; *p* (95% CI)
Diabetes mellitus; n (%)	67 (13.32)	65 (12.48)	58 (12.18)	0.54 (−2.344% to 4.878%	149 (9.30)	0.41 (−1.363% to 3.364%
Hypertension; n (%)	51 (10.14)	46 (8.83)	40 (8.40)	0.33 (−1.479% to 4.887%	96 (6.4)	0.08; (−0.209% to 4.114%)
Dyslipidemia; n (%)	50 (9.94)	49 (9.4)	43 (9.03)	0.64 (−2.292% to 4.097%0	83 (5.53)	0.06; (−0.063% to 4.328%)
Overweight/Obesity; n (%)	49 (9.74)	58 (11.13)	55 (11.55)	0.35 (−1.934% to 4.607%)	265 (17.7)	*p* < 0.0001 (17.303% to 22.946%)
Cardiac disease; n (%)	21 (4.17)	18 (3.45)	19 (3.99)	0.62 (−1.447% to 2.865%)	46 (3.07)	0.002 **; (0.9527% to 4.1357%)
Chronic liver disease; n (%)	12 (2.34)	9 (1.73)	13 (2.73)	0.89 (−1.391% to 1.992%)	34 (2.30)	<0.0001 **; (1.747% to 4.149%)
Chronic renal disease; n (%)	9 (1.79)	8 (1.54)	6 (1.26)	0.562 (−0.873% to 2.064%	59 (3.90)	<0.0001 **; 2.4695% to 5.0889%
Hemoglobinopathies; n (%)	17 (3.38)	20 (3.84)	18 (3.78)	0.72 (−1.841% to 2.229%	65 (4.33)	0.241; (−0.569% to 2.243%)
Neurologic disorders; n (%)	8 (1.59)	10 (1.92)	7 (1.47)	0.88 (−1.534% to 1.386%	57 (3.80)	<0.0001 **; (2.152% to 4.779%)
Autoimmune disorders; n (%)	16 (3.18)	21 (4.03)	19 (3.99)	0.42 (−1.362% to 2.686%)	68 (4.53)	0.473; (−0.929% to 1.997%)
Malignancy (patients with breast cancer, lymphoma, and prostate cancer); n (%)	6 (1.19)	11 (2.11)	8 (1.68)	0.32 (−0.841% to 1.93%	103 (6.87)	<0.0001 **; (3.789% to 6.692%)

Among the enrolled subjects, 1114 (74.27%) vaccinated participants reported no diseases or health disorders; 386 (25.73%) participants had one or more comorbidities, for example, diabetes and hypertension, liver or renal disease, or overweight. Significant. ** Highly significant.

**Table 2 viruses-15-00326-t002:** RT-PCR confirmed COVID-19 cases and COVID-19-related hospitalizations after vaccines and booster doses.

COVID-19 Cases and Hospitalizations after Vaccines and Booster Doses	Group A (n = 503) ChAdOx1 nCoV-19Vaccine	Group B (n = 521) BNT162b2 Vaccine	Group C (N = 476) mRNA-1273	*p* Value and 95% CI between the Three Vaccines	COVID-19 Cases and Hospitalizations in the Control Group throughout the Study (n = 1500)	*p* Value between Vaccinated and Non-Vaccinated Control Subjects
Participants with confirmed RT-PCR COVID-19 (Cases) after the first vaccine dose; (n,%)	113 (22.46%)	66 (12.67%)	67(14.08%)	<0.0001 **	564 (38%)	<0.0001 **
Participants with confirmed RT-PCR COVID-19 (cases) after the second vaccine dose (n,%)	23 (4.6)	16 (3.07)	18 (3. 80)	0.37 (−1.118% to 3.5%)		
Relative risk (RR); (95% CI)	0.124 (0.083 to 0.186)	0.083 (0.051 to 0.135)	0.10 (0.065 to 0.162)			
Vaccine effectiveness (VE); Relative risk reduction (RRR)	89% 0.89	92% 0.92	90% 0.90	ChAdOx1 nCoV-19 vs. BNT162b2: 0.025 ChAdOx1 nCoV-19 vs. mRNA-1273 : 0.298 BNT162b2 vs. mRNA-1273: 0.244		
Absolute risk reduction % (ARR)	37.36%;	38.75%;	38.15%;	0.68 (−4.14% to 6.22%)		
Number needed to vaccinate (NNV) (benefit)	3.09	2.95	3.02			
95% CI	2.724 (Benefit) to 3.570 (Benefit)	2.625 (Benefit) to 3.375 (Benefit)	2.660 (Benefit) to 3.483 (Benefit)			
COVID-19 (cases after the first booster dose; (n,%)	4 (0.79)	3 (0.58)	3 (0.63)	0.67 (−0.664% to 1.461%)		
Relative risk after the first booster (95% CI)	0.024 (0.009 to 0.057)	0.014; (0.0044 to 0.0425)	0.015; (0.0049 to 0.047)	0.19; (−0.444% to 2.784%)		
Relative risk reduction (RRR)	0.98	0.99	0.99	0.51; −(0.844% to 2.158%)		
Absolute risk reduction (ARR)	41.10%	41.40%	41.30%	0.93; −(5.046% to 5.474%)		
Participants with confirmed RT-PCR COVID-19 (Cases) after the second booster dose; (n,%)	0	0	0			
Hospitalizations after the vaccine’s first dose	12 (2.38)	5 (0.95)	7 (1.47)	0.09 (−0.255% to 3.206%)	103/554 (18.6%)	<0.0001
Hospitalizations after the vaccine’s second dose	2 (3.9)	1 (0.19)	0	0.05 *		
Hospitalizations after vaccine boosters 1 or 2	0	0	0			

The relative risk (RR), its standard error, 95% confidence interval, relative risk, absolute risk reduction, and number needed to vaccinate (NNV) is calculated according to Altman, 1991. * Significant. ** Highly significant.

**Table 3 viruses-15-00326-t003:** Vaccine effectiveness in vaccinated participants with comorbidities and COVID-19 incidence in unvaccinated control subjects.

Vaccinated Participants with Comorbidities: (n = 386; 25.73% of the Study Cohort)	COVID-19 Cases after Vaccine Second Dose in Vaccinated Participants with Comorbidities; n (%)	COVID-19 cases after Booster 1	COVID-19-Related Hospitalizations after Vaccines Second Dose or Booster 1	Relative Risk (RR.) (95% CI)	Vaccines Effectiveness in Participants with Comorbidities	COVID-19 Cases in Unvaccinated Controls	*p* Value between COVID-19 Cases in Vaccinated and Unvaccinated Participants with Comorbidities
Diabetes mellitus; n = 190	11/190 (5.8%)	2 (1.05)	1 (0.5%)	0.137 (0.075 to 0.25)	86.3%	63/149 (35.2%)	0.0001
Hypertension; n = 137	2 (1.5%)	0	0	0.04 (0.009 0.163)	96%	35/96 (21.7%)	0.0001
Dyslipidemia; n = 142	1 (0.7%)	0	0	0.019 (0.0054 to 0.251	98%	31/83 (17.9%)	<0.0001
Overweight/Obesity; n = 162 BMI 30–40 BMI > 40	3 (1.85) 17 (10.49)	0 2 (1.23%)	0 0	0.285 (0.186 to 0.439)	82%	166/265 (27.1%)	<0.0001
Cardiac disease; n = 58	0	0	0	0.017 (0.003 to 0.649)	98.3%	23/46 (24%)	0.004**
Chronic liver disease; N = 34	0	0	0	0.04 (0.0052 to 1.218)	98%	12/34 (18.75%)	0.02*
Chronic renal disease; n = 23	1 (4.3%)	1 (4.3%)	0	0.066 (0.009 to 0.451)	93.4%	39/59 (48.1%)	0.005**
Hemoglobinopathies; n = 55	2 (3.6%)	0	0	0.074 (0.019 to 0.294)	92.6%	32/65 (32.3%)	0.002**
Neurologic/psychiatric disorders; n = 25	0	0	0	0.057 (0.004 to 0.912)	94.3%	19/57 (24. 5%)	0.04*
Autoimmune disorders; n = 56	10 (17.9%)	3 (5.4%)	1 (1.79%)	0.229 (0.129 to 0.408)	80%	53/68 (78%)	<0.0001**
Malignancy; n = 25	4 (16%)	2 (8%)	1 (4%)	0.204 (0.082 to 0.502)	80%	81/103 (71.8%)	0.0006**

Several vaccinated and unvaccinated participants had more than one comorbidity. The relative risk (RR), standard error, and 95% confidence interval are calculated (Altman, 1991). * Significant. ** Highly significant.

**Table 4 viruses-15-00326-t004:** Adverse events after the different COVID-19 vaccines.

Parameters	Group A (n = 503) ChAdOx1 nCoV-19 Vaccine	Group B (n = 521) BNT162b2 Vaccine	Group C (n = 476) mRNA-1273 Vaccine	*p* Value; (95% CI)
Pain at the injection site	526 (89.31)	375 (89.07)	308 (90.59)	0.880; (−3.413% to 4.525%)
Fever > 38.5 °C	137 (23.26)	46 (10.93)	57 (16.76)	<0.0001 **; (5.0826% to 13.5058%)
Fatigue > 2 days	294 (49.92)	113 (26.84)	127 (37.35)	*p* < 0.0001 **; (12.2457% to 22.6466%)
Headache > 2 days	174 (29.54)	72 (17.10)	69 (20.29)	*p* < 0.0001 **; (6.8979% to 16.1204%)
Chills	35 (5.94)	19 (4.51)	24 (7.06)	0.975; (−2.474% to 2.690%)
Muscle pain	78 (13.24)	43 (10.21)	31 (9.12)	0.084; (−0.406% to 6.534)
Skin reaction at the injection site; n (%)	45 (7.64)	18 (4.28)	21 (6.1)	0.11; (−0.493% to 4.959%)
Vomiting or diarrhea	13 (2.21)	7 (1.66)	8 (2.35)	0.895 (−1.407% to 1.762%)
Abdominal pain	31 (5.26)	10 (2.31)	13 (3.82)	0.003 **; (0.969% to 5.280%)
Thrombosis	11	1	1	
Numbness	17 (2.89)	4 (0.95)	3 (0.88)	0.005 **; (0.595% to 3.723%)
Joint pains	29 (4.92)	9 (2.14)	7 (2.06)	<0.0001 **; (2.339% to 6.181%)
Sleep disorders	18 (3.06)	3 (0.71)	2 (0.59)	0.457; (−1.091% to 2.591%)
Anxiety/restlessness	5 (0.85)	1 (0.24)	1 (0.29)	0.1020; (−0.194% to 1.765%)
Palpitations; n (%)	17 (2.89)	6 (1.43)	4 (1.18)	0.026 *; (0.1823% to 3.4698%)
Thrombosis	3 (1.8)	0	0	0.013 *; (0.32% to 3.19%)
Myocarditis	1	1	0	1.0000 (−1.4% to 1.4%)
Visual disturbances	1 (0.34)	0	0 (0.00)	1.0000; (−1.1129% to 0.855%)
Anaphylaxis	0	1	1	1.0000 (−1.4% to 1.5%)
Guillain–Barré syndrome	0	1	0	1.0000 (−1.4% to 1.4%)
Seizures	0	0	1	

* Significant. ** Highly significant.

## Data Availability

Not application.

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
