# Peer review of "The Outcome of BNT162b2, ChAdOx1-Sand mRNA-1273 Vaccines and Two Boosters: A Prospective Longitudinal Real-World Study"

_viruses, 2023, doi:10.3390/v15020326_

Round 1
Reviewer 1 Report
The paper describes the study conducted at Prince Sattam Bin Abdulaziz University Hospital and Vaccination Center, Kharj, Kingdom of Saudi Arabia, between March 2021 and September 2022. They assessed the effectiveness of three different vaccines, from Pfizer, Moderna, and AstraZeneca respectively. The data is convincing. Compared to the unvaccinated group, the vaccinated groups with one dose of one of vaccines, showed minor protection but significant protection of confirmed COVID-19 cases, after the boost dose, all three vaccines 89%-92% protection. Then the authors evaluated the AEs of the three vaccines and the dynamics of SARs-CoV-2 antibodies.
Major comments:
1. Figure 2 is confusing. The y axis title is missing. The figure needs legend to explain the data presentation.
2. Please provide title for the table below Figure 2. If the table is included in Figure 2, please give more explanation.
3. Figure 3: please check the label of Figure 3a X axis and the declined time is between 24 to 32 weeks (28.50±3.48 weeks) or as stated in the abstract “24±32 weeks”(line 22).
Please provide a title description for Figure 3 and a figure legend to explain the figure content.
Minor:
1. Please check the font size of the paper. i.e.: Line 340, line 420-423, line 445-453, line 457-459
2. Line 98 “the”
3. Line 142 “management:-“
4. Please check font style of line 152, line 161 and line 166.
5. Please keep the decimal numbers consistent in all the tables.
6. Please keep the font style of figures and tables in the text content consistent.
7. Line 263: “as shown in Table 1”, is that Table 2?
8. Line 331:” as shown in Figure 2”, is that Figure 3?
Author Response
Viruses-2114417 R1
The Outcome of BNT162b2, ChAdOx1-Sand mRNA-1273 Vaccines and Two Boosters: A Prospective longitudinal Real-world Study
Kamal et al
Response to Comments of Reviewer 1
On behalf of all authors, I’d like to thank the reviewer for the constructive comments that aim to strengthen the manuscript. The authors addressed the comments and conducted the required modifications that are underlined.
Major comments:
- Comment: Figure 2 is confusing. The y axis title is missing. The figure needs legend to explain the data presentation.
Response: Thank you for the comment. It seems that the y axis title was truncated during PDF conversion. The figure was inserted. The figure legend is beneath the figure. The legend explains the findings in the figure and the related table.
- Comment: Please provide title for the table below Figure 2. If the table is included in Figure 2, please give more explanation.
Response: The table below figure 2 is related to the figure. The legend explains the findings in the figure and the related table.
- Figure 3: please check the label of Figure 3a X axis and the declined time is between 24 to 32 weeks (28.50±3.48 weeks) or as stated in the abstract “24±32 weeks”(line 22).
Response: Thank you for bringing the typo to our attention. The mistake was corrected in the abstract, stating that the titers declined after a mean of 28.50±3.48 weeks.
- Comment: Please provide a title description for Figure 3 and a figure legend to explain the figure content.
Response: The titles of figures 3 a and b are shown above the figures. The figure legends exist below the figures and include explanations to the figures
Minor:
- Comment: Please check the font size of the paper. i.e.: Line 340, line 420-423, line 445-453, line 457-459
Response: The fonts and formatting are adjusted
- Comment: Line 98 “the”
Response: Fixed
- Line 142 “management: -“
Response: Fixed
- Please check font style of line 152, line 161 and line 166.
Response: All fonts have been adjusted to Times New Roman 12
- Please keep the decimal numbers consistent in all the tables.
Response: the decimal spaces were adjusted
- Please keep the font style of figures and tables in the text content consistent.
Response: All fonts have been adjusted to Times New Roman 12
- Line 263: “as shown in Table 1”, is that Table 2?
Response: The description of the malignancies is shown in table 1
- Line 331:” as shown in Figure 2”, is that Figure 3?
Response: Figure 2 is a Kaplan Meier graph that demonstrates the time to event (SARS-CoV2 infection)
Figure 3 shows the kinetics of the antibody responses.
Both figures were associated with legends and the findings are explained.

Reviewer 2 Report
No comments
Author Response
Thank you very much for reviewing the manuscript. We appreciate your time and effort.
Reviewer 3 Report
The article is a well-written, real world, prospective cohort study about COVID-19 vaccine effectiveness and safety. Before considering it for publication, however, few major points should be addressed:
- I would avoid vaccine commercial names.
- How many patients did you screen to gather 1500 people to whom vaccine was contra-indicated? The total number of eligible patients should be clarified. In particular, it is unclear whether the total number if patients considered for the study were 3000 (503+521+476+1500). In this case, it seems unlikely that 50% of the sample presented contra-indications to all vaccines. Also, vaccines included in the study have different contra-indications. This should be clarified.
- Authors should provide more detail about follow-up. How long were patients followed? How were follow-up visit structured? How many follow-up visits were done for each patient?
-
- Also, it is my understanding that patients were tested only if they self-reported symptoms. Besides the limitation of missing the asymptomatic cases, that was well-acknowledged, patient-initiated reporting may have lead to further missing cases of COVID-19. Were patients actively followed or results rely only on patient reporting? This should be clarified and this limitation should be acknowledged, especially in consideration that incidence of COVID was reported to be one of the outcomes of the study.
Author Response
Viruses-2114417 R1
The Outcome of BNT162b2, ChAdOx1-Sand mRNA-1273 Vaccines and Two Boosters: A Prospective longitudinal Real-world Study
Kamal et al
Response to Comments of Reviewer 3
On behalf of all authors, I’d like to thank the reviewer for the constructive comments that aim to strengthen the manuscript. The authors addressed the comments and conducted the required modifications that are underlined.
Major comments:
Comment: - I would avoid commercial vaccine names.
Response: Thank you for the comment. We revised the MS and replaced the commercial vaccine names with generic ones. However, the commercial names were mentioned once in the introduction to familiarize the reader with the vaccine name just in case some readers might not recognize the generic names.
Comment: - How many patients did you screen to gather 1500 people to whom vaccine was contra-indicated? The total number of eligible patients should be clarified. In particular, it is unclear whether the total number if patients considered for the study were 3000 (503+521+476+1500). In this case, it seems unlikely that 50% of the sample presented contra-indications to all vaccines. Also, vaccines included in the study have different contra-indications. This should be clarified.
Response: As shown in the results and Figure 1, initially, 4145 vaccinees were screened for eligibility for the study. Three thousand five hundred and thirty-five were eligible; however, only 1500 subjects agreed to join the study, provided informed consent, and were enrolled. Six hundred and ten (14%) were ineligible due to several contraindications such as a history of severe allergic reactions, myocarditis or pericarditis, acute decompensated heart failure, history of cerebral venous sinus thrombosis (CVST), heparin-induced thrombocytopenia (HIT), idiopathic splanchnic (mesenteric, portal, splenic) thrombosis, antiphospholipid syndrome with thrombosis agreed to join the study as control subjects. In addition, at the early stage of the vaccine program in KSA, some groups had temporary vaccine exemptions, such as patients recovering from natural COVID-19, pregnant or lactating women, patients treated with immunosuppressive therapies etc. Thus, 890 subjects with temporary vaccine exemptions were also enrolled and followed as controls in addition to the 610 ineligible patients (890+610=1500).
Comment- Authors should provide more detail about follow-up. How long were patients followed? How were follow-up visits structured? How many follow-up visits were done for each patient?
Response: The flow of participants through the study and follow-up is shown on page three (lines 126-130) and in Figure 1.
Comment: Also, it is my understanding that patients were tested only if they self-reported symptoms. Besides the limitation of missing the asymptomatic cases, that was well-acknowledged, patient-initiated reporting may have lead to further missing cases of COVID-19. Were patients actively followed or results rely only on patient reporting? This should be clarified and this limitation should be acknowledged, especially considering that incidence of COVID was reported to be one of the study's outcomes.
Response: Participants were advised to report to the hospital to get tested for COVID-19 using RT-PCR or COVID-19 antigen when they developed any mild symptoms or were in contact with COVID-19-positive patients. Participants who tested positive were considered "cases." In addition, COVID-19-related hospitalizations for participants were monitored and reported. We agree that patient reporting may miss some COVID-19 cases. We addressed this limitation in page 20.

Round 2
Reviewer 3 Report
Results presentation improved and limitations were properly acknowledged. However, it is still unclear how the control population was selected, since contraindications are vaccine-specific. For example, if a subject suffers from recurrent thrombosis that contraindicates the administration of a viral vector vaccine, he/she might always receive a mRNA vaccine. Also, the "temporary exemption" should be further clarified: how many patients were temporarily exempted and then received vaccination?
Author Response
Response to Reviewer's 3 Comments
Comment: Results presentation improved, and limitations were adequately acknowledged.
Response: Thanks for the reviewer's comments that helped to strengthen the manuscript. The authors addressed the comments regarding the control group below.
Comment: However, it is still unclear how the control population was selected since contraindications are vaccine-specific. For example, if a subject suffers from recurrent thrombosis that contraindicates the administration of a viral vector vaccine, he/she might always receive an mRNA vaccine.
Response:
Thank you for the comment regarding the characteristics of the control group. Therefore, more details and data have been added and underlined on pages 4 and 7. We want to clarify the following:
- The current study was conducted in a real-life setting and started at an early phase of COVID-19 vaccination and extended for 18 months, during which there were variations in the approved vaccines and their availability, the vaccination criteria, the vaccine prioritization policies etc
- Although the BNT162b2 vaccine was the first COVID-19 vaccine licensed and administered in KSA (mid-December 2020), the limited vaccine supply resulted in prioritizing vaccination of high-risk groups such as frontline healthcare workers, healthcare professionals at risk of infection, old individuals (>65 years), and those living in nursing homes. The vaccination started in major cities (Riyadh, Jeddah, Dammam, Madinah, and Makkah) where vaccination was performed in centers with ultra-cold storage and transport capabilities in five major Saudi cities and prioritized
- ChAdOx1 nCoV-19 was then licensed in February 2021. Due to the flexibility in storage and handling of the ChAdOx1 nCoV-19 vaccine, it was easy to distribute the vaccine to more regions of the Kingdom.
- The current study was conducted in a University Hospital vaccination center that started COVID-19 in March 2021. Initially, the ChAdOx1 nCoV-19 vaccine was the only vaccine available in the center. BNT162b2 and mRNA-1273 were available in the center in August 2021 and November 2021, respectively. Since no alternative vaccine was available in the vaccination center then, patients with contraindications to ChAdOx1 nCoV-19 vaccine were on a waiting list to receive mRNA vaccines when available.
- The control group consists of individuals with contraindications to either vector vaccines or mRNA vaccines (in our center persons, some persons had an allergy to components in both vaccines, though few) or persons with a transient medical condition that warrants delaying the vaccination (temporary exemption from vaccinations; explained later) or the presence of contraindication to a specific vaccine with the unavailability of an alternative vaccine.
- Subjects with a temporary exemption from vaccination were enrolled in the study if they provided informed consent. They were followed until the reason for exemption was resolved and then vaccinated.
- Comment: Also, the "temporary exemption" should be further clarified: how many patients were temporarily exempted and then received vaccination?
- Response: Temporary exemption is postponing or deferring the vaccine for a specific time interval due to a medical condition approved by the Public Health Authorities. The definition of temporary exemption was added, and the causes of postponing vaccination and the number of individuals who received the vaccine after the cause of temporary exemption were resolved were shown.
Conditions eligible for temporary COVID-19 vaccine exemption include the following:
- Inflammatory cardiac illness within the past three months; myocarditis or pericarditis; acute rheumatic fever or acute rheumatic heart disease (i.e., with active myocardial inflammation); or acute decompensated heart failure
- Vaccination was temporarily deferred up to 4 months after a confirmed infection with COVID-19
- Until recovery from a severe medical condition (such as major surgery or hospital admission for a serious illness)
- The serious adverse event caused by a previous dose of a COVID-19 vaccine, with no acceptable alternative vaccine available
- Treatment with a monoclonal antibody for COVID-19
- Initially, data were limited about the vaccine's safety in immunocompromised patients (HIV, patients on immunosuppressant drugs, and immunocompromising diseases). The treating physician and the patient discuss the available safety data before administering the vaccine to evaluate risks and benefits.
- In the early phase of the study, data were limited about the vaccine's safety in pregnant women. Thus, pregnant women, particularly those with high-risk pregnancies, were temporarily exempted until delivery.
The temporary exemption procedure is as follows:
- The health authorities issue the decision of temporary exemption upon submitting the patient's medical report and investigations by the treating physician or hospital on a specific platform linked to the MoH.
- A decision of exemption or taking the vaccine is made in two to five days.
- If temporary vaccine exemptions are granted, the subject's vaccination status is designated on the electronic vaccination platform, and the mobile application as 'Temporary Exempted from COVID-19 Vaccination". In addition, the duration of the exemption is specified.
- At the end of the temporary exemption period, the subject is re-evaluated for vaccine eligibility and then provided the vaccine.